# Structure and Property of Diamond-like Carbon Coating with Si and O Co-Doping Deposited by Reactive Magnetron Sputtering

Wei Dai *, Liang Wu and Qimin Wang

School of Electromechanical Engineering, Guangdong University of Technology, Guangzhou 510006, China
* Correspondence: weidai@gdut.edu.cn; Tel.: +86-13022097190

**Abstract:** In this paper, diamond-like carbon (DLC) coatings with Si and O co-doping (Si/O-DLC) were deposited by reactive magnetron sputtering using a gas mixture of $C_2H_2$, $O_2$ and Ar to sputter a silicon/graphite splicing target. The O content in the Si/O-DLC coatings was controlled by tuning the $O_2$ flux in the gas mixture. The composition, chemical bond structure, mechanical properties and tribological behavior of the coatings were investigated by X-ray photoelectron spectroscopy, Fourier infrared spectrometer, Raman spectroscopy, nanoindentation, a scratch tester and a ball-on-disk tribometer. The electrical resistivity of the Si/O-DLC coatings was also studied using the four-point probe method. The results show that the doping O tends to bond with Si to form a silicon–oxygen compound, causing a decrease in the $sp^3$ content as well as the hardness of the coatings. The tribological performance of the coatings can be improved due to the formation of the silicon–oxygen compound, which can effectively reduce the friction coefficient. In addition, the insulating silicon–oxygen compound is doped into the C-C network structure, significantly improving the surface resistivity of the DLC coating with a low $sp^3$ content. The Si/O-DLC coatings with good mechanical properties, tribological performance and electrical insulation properties might be used as protection and insulation layers for microelectronics.

**Keywords:** diamond-like carbon; doping; mechanical property; tribology; resistivity

## 1. Introduction

Diamond-like carbon (DLC) coatings have attracted considerable attention due to their unique combination of desirable properties, including a high hardness, low friction, chemical inertness and high wear resistance [1–3]. Recently, many elements, including metallic elements (Cr [4], Ti [5], W [6] et al.) and non-metallic elements (Si [7], O [8], N [9] et al.), have been doped into DLC to improve the coating properties, such as the residual stress, toughness, wettability, electrical properties and tribological behavior. According to the chemical property and content of the doping element, the doping atoms can bond with carbon (carbide elements, such as Cr, Ti, W, N and Si) and exist as a part of the carbon matrix, or can create a two-dimensional array of nano-clusters (carbide nanoparticles, metallic precipitated phase et al.) within the DLC matrix or an atomic-scale composite dissolving in the DLC matrix [4–9]. It is clear that the chemical state and existing form of the doping atoms would pronouncedly influence the properties of the DLC coatings. Among these elements, the incorporated Si can substitute carbon atoms in the carbon network and maintain the $sp^3$-hybridized bond structure since Si does not form $\pi$ bonds, which changes the electronic properties of the DLC coatings [10]. On the other hand, the added O is expected to facilitate the formation of a polycarbonate-like type of structure, causing an increase in the coating optical gap and a decrease in the refractive index [11]. DLC coatings with Si and O co-doping (Si/O—DLC) are also reported. In the Si/O-DLC coating, Si and O are mainly composed of Si-O and Si–C bonds embedded in a carbon network structure, and amorphous $SiO_x$ (a-$SiO_x$) is believed to separate from the carbon network [12]. The separated a-$SiO_x$ phase doped in the C-C network structure can increase the dielectric

constant of DLC and make the coating have good insulation [11]. In addition, the Si-O compound formed on the surface can effectively reduce the friction coefficient of the Si/O-DLC coating [13,14]. The formation of the separated a-SiO$_x$ phase has a significant relationship with the deposition technology and the Si and O doping contents. Randeniya et al. [15] deposited Si/O-DLC by pulsed direct current plasma-enhanced chemical vapor deposition (DC-PECVD) using tetraethoxysilane (($C_2H_5O$)$_4$Si as the precursor, and found that the Si/O–DLC was formed as a single phase with siloxane and O–Si–C$_2$ bonding networks when the Si content was not higher than 13 at.%, while the Si/O–DLC was composed of siloxane bonding networks and a segregated a-SiO$_x$ phase when the Si content was greater than 13 at.%. Damasceno et al. [16] prepared Si/O-DLC films by PECVD using $CH_4$:$SiH_4$:$O_2$ gas mixtures and observed that silicon atoms are incorporated into the DLC phase bonded to carbon at a low $O_2$ flux. However, when the $O_2$ flux is increased, phase segregation is expected to take place, since Si–O bonds are favored compared to Si–C or C–O bonds.

It should be noted that PECVD is frequently used for the deposition of DLC-SiO$_X$ coatings [12]. Special silicon/oxygen-based precursors, including gaseous (silane ($SiH_4$) and oxygen ($O_2$) mix [17]; tetramethyl silane—TMS (($CH_3$)$_3$SiH) and $O_2$) mix [18]) and liquids (hexamethyldisiloxane-HMDSO, $C_6H_{18}OSi_2$) [19,20], are needed in the PECVD process. However, these silicon/oxygen-based precursors are very expensive and the types of precursors are limited.

In this paper, Si/O-DLC coatings were deposited using physical vapor deposition–magnetron sputtering technology, where a gas mixture of $C_2H_2$, $O_2$ and Ar was used to sputter a silicon/graphite splicing target. Magnetron sputtering is expected to have the advantages of a low cost, simple process and little pollution. The relative contents of Si, O and C in the Si/O-DLC coatings were adjusted by changing the $O_2$ flux in the gas mixture. The influence of the $O_2$ flux on the chemical composition, bonding structure, mechanical properties, tribological performance and electrical resistance of the coatings were researched. The relationship between the preparation technology, composition, structure and properties of the coatings were discussed.

## 2. Materials and Methods

Series of Si/O-DLC coatings were deposited on cemented carbide (WC with 6 wt.% Co, mirror polish) substrates, silicon wafers and quartz glass sheets using DC magnetron sputtering (DCMS) unit equipped with a graphite–silicon splicing plane target with length of 470 mm and width of 210 mm (as shown in Figure 1b). The distance between the sputtering target and the substrate holder was approximately 50 mm. A linear ion source (LIS) device was situated opposite the DCMS unit (as shown in Figure 1a). The substrate holders were installed on the central rotary table, which revolved constantly at a speed of 2 rpm. Each of the substrates were cleaned sequentially in ultrasonic baths of acetone and alcohol for 20 min, and were then put into the vacuum chamber after drying in the thermostat oven. The base pressure in the vacuum chamber was below $2 \times 10^{-3}$ Pa. Ar glow discharge cleaning with −650 V bias voltage was carried out for 20 min to clean the substrates and the chamber. Then, 100 sccm Ar was input into the LIS to produce Ar$^+$ beam, which was used to etch the sample for 20 min to remove impurities and oxide on the substrate surface before deposition. The current of the LIS was 2.5 A and the bias was −700 V. During coating process, gas mixture of $C_2H_2$, Ar and $O_2$ were inputted into chamber. Ar was directly input into the DCMS unit. $C_2H_2$ and $O_2$ were input through other inlets to avoid the target "poisoning". The $C_2H_2$ flux and Ar flux were set at 30 sccm and 170 sccm, respectively. The $O_2$ flux ranged from 0 to 6 sccm in the precursor gases. The working pressure was kept at around 0.5 Pa and the DC power was approximately 2 kW. The bias voltage was −100 V and the deposition duration was 90 min.

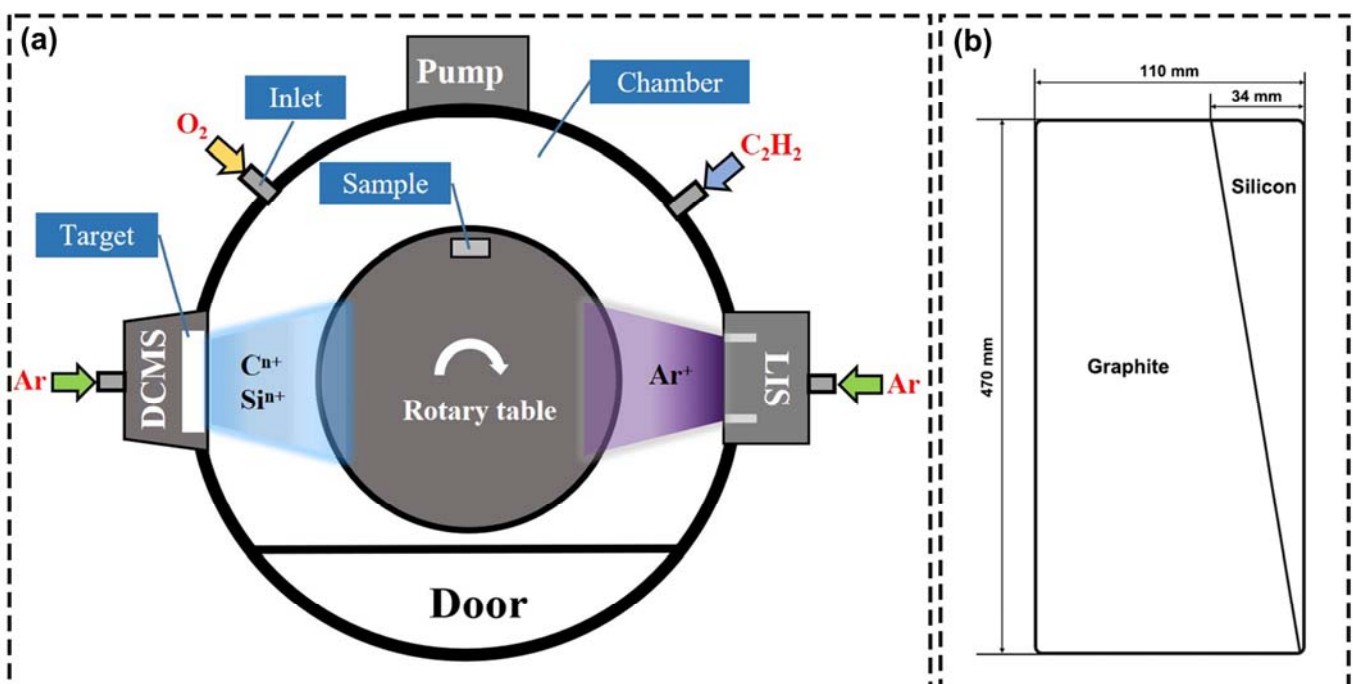

**Figure 1.** Schematic diagrams of (**a**) deposition chamber and (**b**) the graphite–silicon splicing target.

The thickness of the deposited coatings was measured using ball crater tester (CaloTest CAT$^2$c, Anton Paar, Austria) and ranged from approximately 0.7 μm to 0.6 μm as the $O_2$ flux increased from 0 sccm to 6 sccm. The elemental compositions and chemical bonds of the as-deposited coatings were characterized using an X-ray photoelectron spectroscopy (XPS, Thermo ESCALAB 250Xi) with Al (mono) Kα (hv = 1486.6 eV), with a step size of 1 eV. The high-resolution XPS spectra were recorded at pass energy of 100 eV with a step size of 0.05 eV. The typical C-C bond around the binding energy of 284.6 eV [21] in C1s spectra of the hydrogenated DLC was used for the calibration. Peak fitting was performed using the CasaXPS software. Shirley-type background and asymmetric Lorentzian/Gaussian (70%/30%) peak shapes were used. Before measurement, all of the samples were etched by $Ar^+$ ion beam with energy of 2 kV for 5 min to remove contaminants. The etch depth was approximately 10 nm. The contents of the elements in the coatings were also calculated according to the atomic sensitivity factors and the relative area ratios of the peaks in XPS spectra of the coatings. Transmission Fourier transform infrared spectroscopy (FTIR) (Nicolet iS10, Thermo Fisher Scientific Inc., Waltham, MA, USA) was used to detect the chemical structure changes of the coatings deposited on quartz glass sheets. The collection parameters were set as follows: resolution of 4 cm$^{-1}$ and wave number range of 4000 cm$^{-1}$ to 400 cm$^{-1}$. A total of 32 scans were performed and averaged per spectrum to improve the signal-to-noise ratio. The carbon atomic bonding structure of the coatings was characterized using Raman spectroscopy (LabRAM HR Evolution, Irvine, CA, USA) with incident light from a He-Ne laser at a wavelength of 632.8 nm. The laser output was approximately 0.8 mW. A neutral density filter was used to reduce the laser intensity and only 10% laser power was used to detect the sample. The measurement positions were focused at a magnification of 100× in the optical microscope and a light spot diameter ≤1 um. The Raman scattering range spanned from 600 cm$^{-1}$ to 2000 cm$^{-1}$. The spectral resolution was approximately 1 cm$^{-1}$.

The coating hardness and elastic modulus were measured using a nanoindentation tester (CSM, TTX-NHT) with a Berkovich diamond indenter under constant load of 5 mN. In order to minimize the influence of the soft substrate on the hardness measurement, the indentation depths were controlled to approximately 10% of the coating thicknesses. Fifteen replicate indentations were made for each sample. The adhesion strength of the coat-

ings on cemented carbide substrates was tested by a micro-Scratch Tester (CSM, Revetest scratch tester) using Rockwell C diamond styli with radius of 200 um. A normal load range of 1 to 100 N, scratch length of 3 mm and scratch speed of 6 mm/min were used in the experiments. Post-test characterization of the scratch-tested samples was performed through observations under optical microscopy. The tribological behaviors of the coatings were measured using a ball-on-plate tribometer (CSM, THT) with an $Al_2O_3$ ball (6 mm in diameter) as a counterpart material. The sliding tests were conducted with a rotational speed of 800 r/min under a load of 2 N (the initial Hertzian contact pressure was approximately 0.2 GPa) at ambient air condition (relative humidity: $50 \pm 5\%$, RT), and the total wear duration was 5000 cycles. The electrical resistivity of the coatings was measured using four-probe method at room temperature. The model of the resistance meter used was Merrick RK2514 precision resistance tester. The resistance test range was 0.1 $\mu\Omega$~110 M$\Omega$, and the test accuracy was 0.01%.

## 3. Results

A high-resolution XPS test was processed to characterize the chemical states of the Si/O-DLC coatings. Figure 2 shows the typical high-resolution XPS spectra for C1s, O1s and Si2p in the Si/O-DLC coatings deposited with the different $O_2$ fluxes. It can be seen that the C1s spectra of the coatings (Figure 2a) can be divided into three peaks. The peak at the binding energy of $283.8 \pm 0.3$ eV can be considered as the Si-C bond, the peak at the binding energy of $284.9 \pm 0.2$ eV can be assigned to the C-C bond and the peak around the binding energy of $286.2 \pm 0.2$ eV can be attributed to the C-O bond [10,14]. In fact, the C-C bond can be divided into an $sp^2$ C-C bond and $sp^3$ C-C bond. However, it is difficult to deconvolve the C-C bond since the binding energy of the $sp^2$ C-C bond (approximately 284.5 eV) is very close to that of the $sp^3$ C-C bond (approximately 285.0 eV). Accordingly, the C-C bond was not deconvolved here. The change in the $sp^2/sp^3$ ratio is discussed via Raman below. It should be noted that the intensity of the C 1s spectra decreases and the peaks broaden as the $O_2$ flux increases, indicating an increase in the fraction of the Si-C bond. For the O1s spectra (Figure 2b), two peaks can be fitted. The peaks located at the binding energies of $531.9 \pm 0.2$ eV and $532.4 \pm 0.2$ eV can be considered as a Si-O bond and C-O bond, respectively [14]. It should also be noted that the intensity of the O1s spectra increases with the $O_2$ flux, implying that the O content in the coatings increases as the $O_2$ flux increases. In addition, the major peak of the O1s is the O-Si bond, meaning that the doped O is inclined to bond with Si.

The Si2p spectra (Figure 2c) can be fitted with two peaks at $100.6 \pm 0.1$ eV and at $101.6 \pm 0.3$ eV, which correspond to the Si-C and Si-O, respectively [12]. It can be seen that the Si-C bond fraction decreases whereas the Si-O fraction increases as the $O_2$ flux increases. Normally, the doping Si is mostly inclined to bond with the $sp^3$-C due to the electron structure [10]. When O was doped into the coating, O first combined with Si to form a Si-O bond since Si-O bonds are favored compared to Si-C or C-O bonds. It is reported that the Si/O-DLC structure is a stable network structure formed by Si-O and a stable network structure formed by C-C/C-H with a weak Si-C bond combination in the middle [22].

The composition variation curves of C, O and Si in the Si/O-DLC coatings prepared with different $O_2$ flow rates are shown in Figure 3. The hydrogen content in the coatings was neglected due to a lack of signal intensity in the current XPS detection measurement. As the $O_2$ flux increases from 0 sccm to 6 sccm, the O content increases from 2.2 at.% to 18.1 at.% whereas the C content decreases from 85.38 at.% to 69.65 at.%. The Si content firstly slowly decreases from 12.42 at.% to 8.87 at.% as the $O_2$ flux increases from 0 sccm to 4 sccm, and then increases to 12.25 at.% when the $O_2$ flux increases to 6 sccm. It is clear that the incorporation of O results in the decreases in C and Si contents. However, the doping O is more reactive than C and favors combining with Si to generate stable Si-O compounds, which contribute to the increase in the Si content. The presence of oxygen in the coating deposited without $O_2$ (0 sccm) might be due to the residual oxygen in the chamber and the oxide in the target.



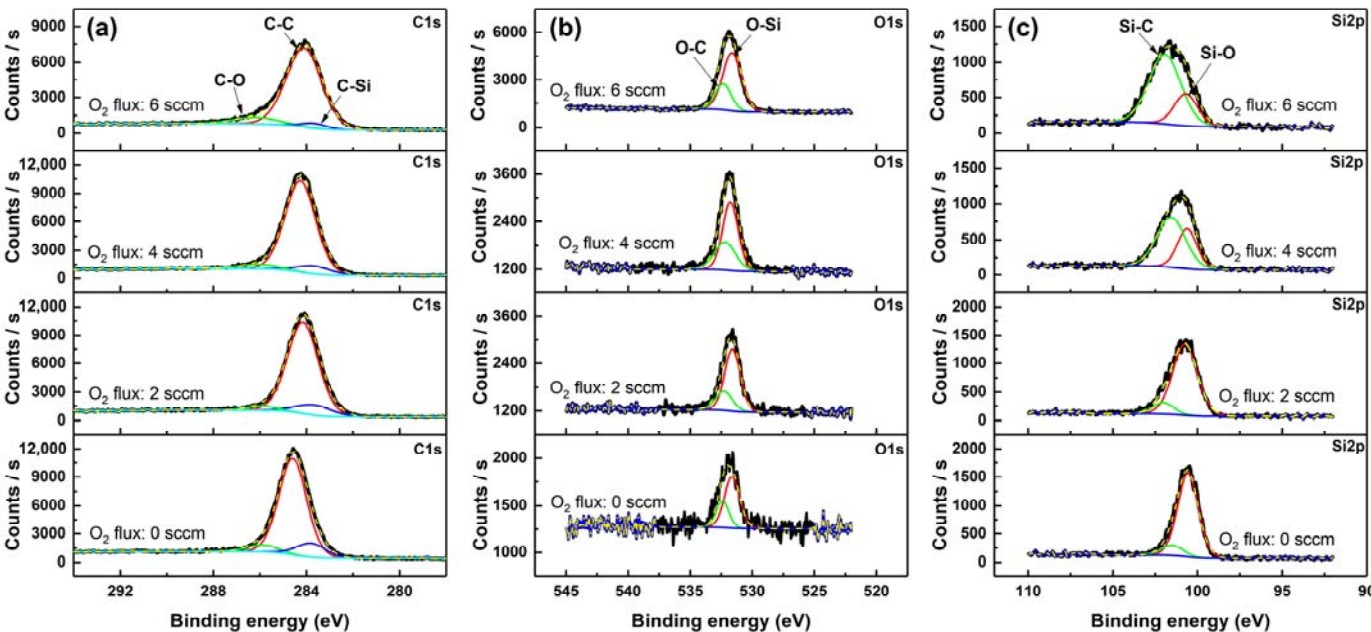

**Figure 2.** High-resolution XPS spectra for (**a**) C1s, (**b**) O1s, and (**c**) Si2p regions for the Si/O-DLC coatings deposited with different $O_2$ fluxes.

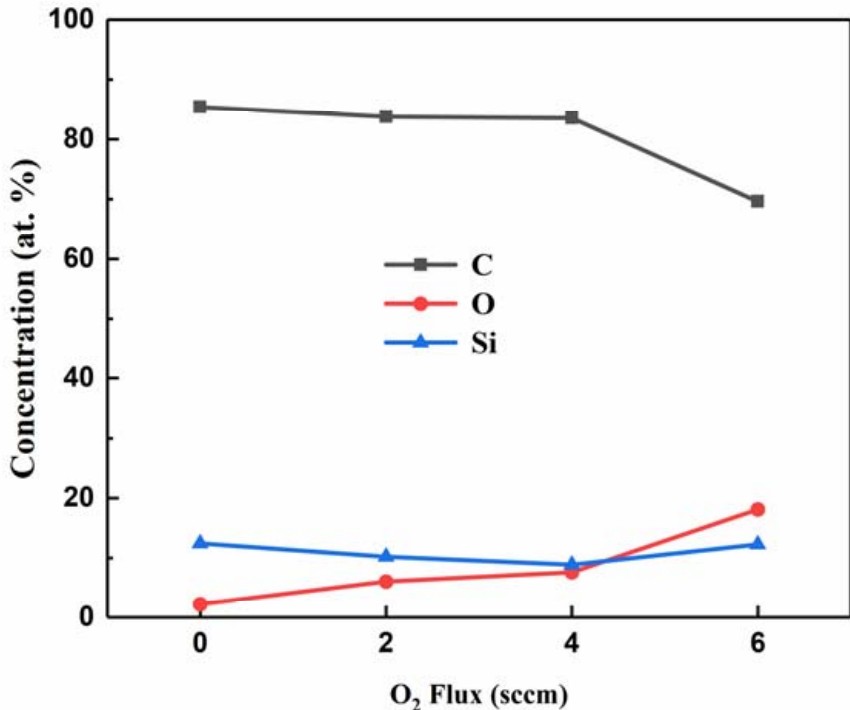

**Figure 3.** Si, O and C contents of the Si/O-DLC coatings detected by XPS as a function of the $O_2$ flux.

Figure 4 reveals the FTIR spectra of the Si/O-DLC coatings deposited with different $O_2$ fluxes. Spectra are shifted vertically one by one for clarity. It can be seen that the main absorption peaks of the Si/O-DLC coatings correspond to the Si-C, Si-O-Si, C=C and $-CH_2$ bond vibrations. Furthermore, the weak peaks assigned to Si-H and $-OH$ bond vibrations are also detected in the FTIR spectra. The absorption peaks of Si-C, Si-O-Si, C=C, $-CH_2$ and Si-H appear at around 827 $cm^{-1}$, 1000 $cm^{-1}$, 1600 $cm^{-1}$, 2918 $cm^{-1}$, and 2173 $cm^{-1}$, respectively [12,23]. A small peak of $-OH$ appears at approximately 3500 $cm^{-1}$ when the $O_2$ flux is 6 sccm. For the Si-DLC coating (the $O_2$ flux is 0 sccm), an obvious Si-H peak can

be found, indicating that Si mainly exists in the form of Si-H and Si-C. Furthermore, the doping Si tends to combine with O when the O is doped. In addition, when the amount of O is sufficient, a small amount of −OH functional groups appear.

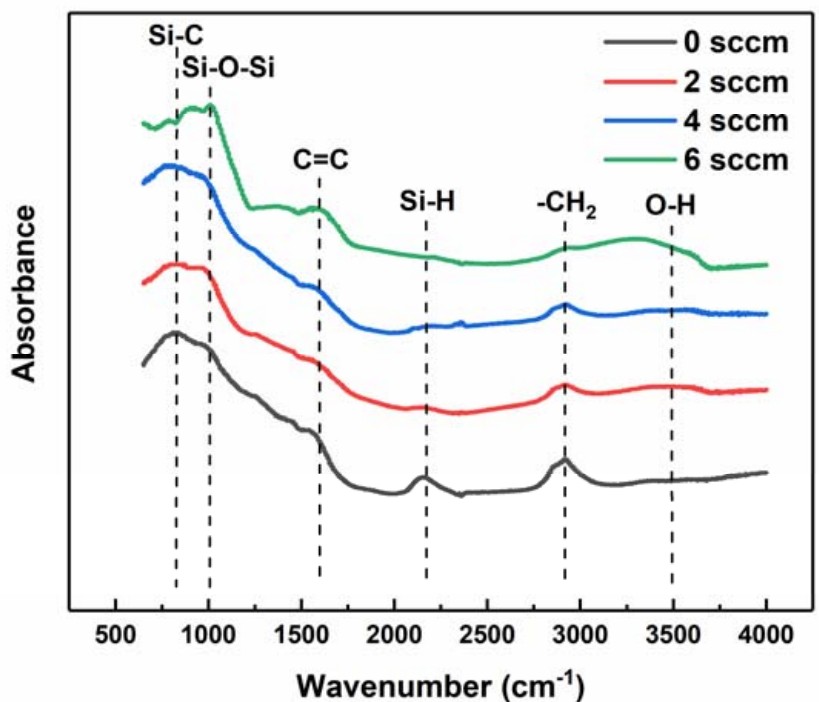

**Figure 4.** FTIR spectra of the Si/O-DLC coatings deposited with different $O_2$ fluxes. The thicknesses of the coatings range from approximately 0.7 μm to 0.6 μm as the $O_2$ flux increases from 0 sccm to 6 sccm.

It should be noted that the intensity of the Si-C peak decreases whereas the intensity of the Si-O-Si absorption peak increases when increasing the $O_2$ flux, indicating that the Si-C bond fraction decreases whereas the Si-O-Si fraction increases in the coatings. The intensity of the C=C absorption peak tends to increase with the $O_2$ flux, and exhibits a significant increase when the $O_2$ flux is 6 sccm, implying that the C=C bond fraction increases in the coatings. The increase in the C=C bond fraction demonstrates that the graphitization occurs in the DLC coatings [24]. It is clear that the FTIR results are consistent with the XPS result. According to XPS and FTIR results, the doping O favors reacting with Si to form Si-O-Si bonds, causing the reduction in the fraction of the Si-C bond that is expected to maintain the $sp^3$-hybridized bond structure [10]. As a result, the Si/O-DLC coatings tend to graphitize when increasing the $O_2$ flux.

Raman spectroscopy is a popular and effective tool used to characterize carbon bonding in carbon materials (e.g., DLC). The Raman spectra of the DLC coatings are dominated by the $sp^2$ sites since the π states have a lower energy than the σ states and are thus more polarizable, and the $sp^2$ sites that consist of two π orbits and two σ orbits have a 50–230 times larger Raman cross-section than $sp^3$ sites, which only contain four σ orbits [24]. In order to determine the carbon atomic structure of the Si/O-DLC coatings, the Raman spectra of the coatings were tested and are shown in Figure 5a. A broad asymmetric Raman scattering band ranging from 1000 to 1700 cm$^{-1}$ can be seen in the Raman spectra, which is the typical characteristic of DLC [25]. The asymmetric Raman spectra can be fitted using two Gaussian peaks: the G-peak around 1580 cm$^{-1}$ and the D-peak around 1360 cm$^{-1}$, which originate the vibrations of the C-C stretching of all $sp^2$ pairs and the symmetric breathing vibration of $sp^2$ atoms only in rings [26,27]. The $sp^2/sp^3$ ratios of the DLC coatings can be characterized according to the G-peak position and the intensity ratio of the D-peak to G-peak ($I_D/I_G$). The G-peak shift toward a lower wavenumber and the decrease in the $I_D/I_G$ ratio correspond to the decrease in the $sp^2/sp^3$ ratio [25–27]. Figure 5b shows the

corresponding G-peak position and the $I_D/I_G$ ratio of the coatings after being fitted. It can be seen that the $I_D/I_G$ ratio increases and the G-peak position shifts to a larger wavenumber as the $O_2$ flux increases, implying that the $sp^2/sp^3$ ratio increases. It is clear that the Raman results are consistent with the results of FTIR. The O doping facilitates the graphitization of the coatings.

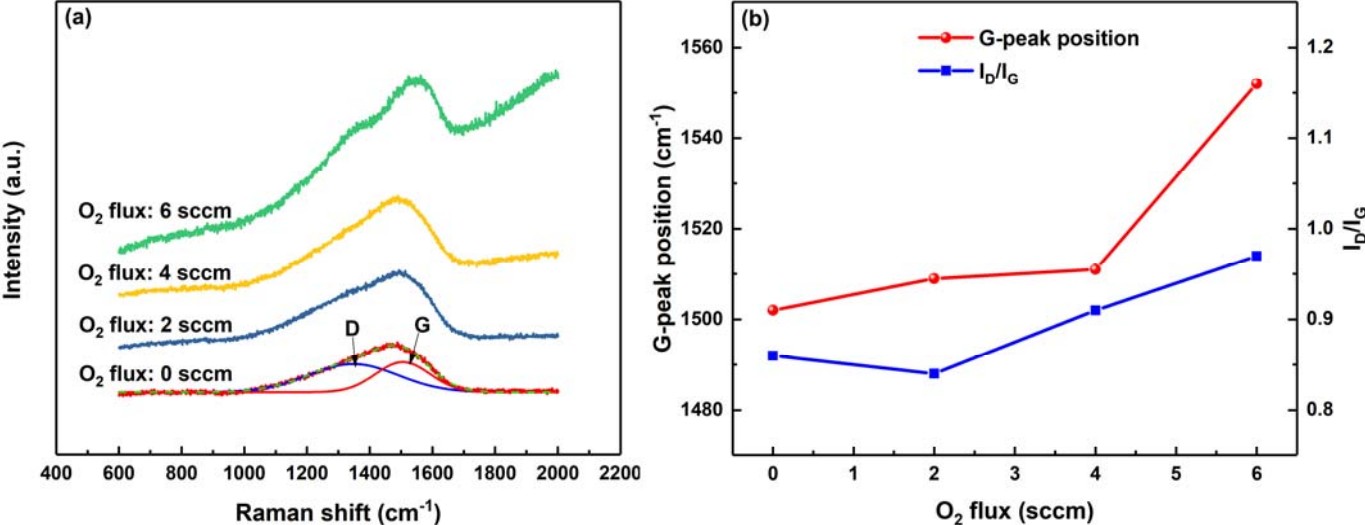

**Figure 5.** (**a**) Raman spectra of the Si/O-DLC coatings deposited with different $O_2$ fluxes and the typical Raman spectrum that is fitted using two Gaussians peaks of G-peak and D-peak; (**b**) G-peak position and $I_D/I_G$ of the Raman spectra of the coatings as a function of the $O_2$ flux.

Figure 6 presents the hardness and elastic modulus of the Si/O-DLC coatings deposited with various $O_2$ fluxes. The hardness and elastic modulus of the pure Si-DLC coating (the $O_2$ flux = 0 sccm) are approximately 10 GPa and 151 GPa, respectively. As a small amount of O is added (the $O_2$ flux = 2 sccm), the hardness of the coatings is almost maintained at approximately 10 GPa. It should be noted that the hardness of the Si/O-DLC coating with a small amount of O doping is comparable to that of $SiO_2$, which is widely used in insulating materials [28]. However, as the $O_2$ flux increases from 2 sccm to 6 sccm, the hardness and elastic modulus of the Si/O-DLC coatings decrease from approximately 10 GPa to 6.6 GPa and from approximately 110 GPa to 58.5 GPa, respectively. The reduction in the hardness and elastic modulus of the Si/O-DLC coatings is believed to mainly be related to the carbon structure. As shown in the results of Raman (Figure 4), the $sp^2/sp^3$ ratio increases with the $O_2$ flux, resulting in a decrease in the mechanical properties of the coatings.

A scratch test was used to characterize the adhesion strength of the Si/O-DLC coatings. The critical load (Lc) when coating exfoliation begins to appear in the scratch is usually used to evaluate the coating–substrate bonding strength. The optical morphology of the scratch tracks of Si/O-DLC coatings deposited with different $O_2$ fluxes is shown in Figure 7. It can be seen that the critical load of the pure Si-DLC is approximately 12 N. When the O is doped into the coatings, the coating critical load sharply increases up to 44 N. This means that the doping O can significantly improve the adhesion of the Si/O-DLC coatings. However, the critical load of the coatings decreases when increasing the $O_2$ flux, and decreases to approximately 8 N when the $O_2$ flux is 6 sccm. The enhancement of the adhesion of the Si/O-DLC might be due to the decrease in the internal stress caused by the introduction of O [29,30]. As the $O_2$ flux increases, the hardness of the coatings decreases, causing a reduction in the scratch resistance.

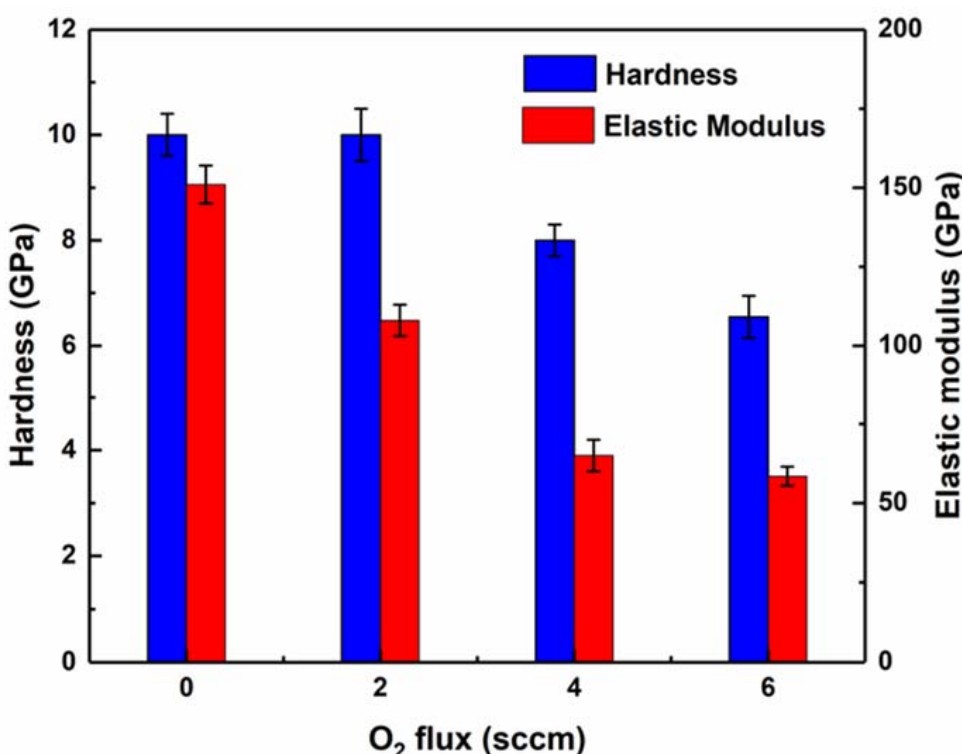

**Figure 6.** Hardness and elastic modulus of the Si/O-DLC coatings deposited with different $O_2$ fluxes.

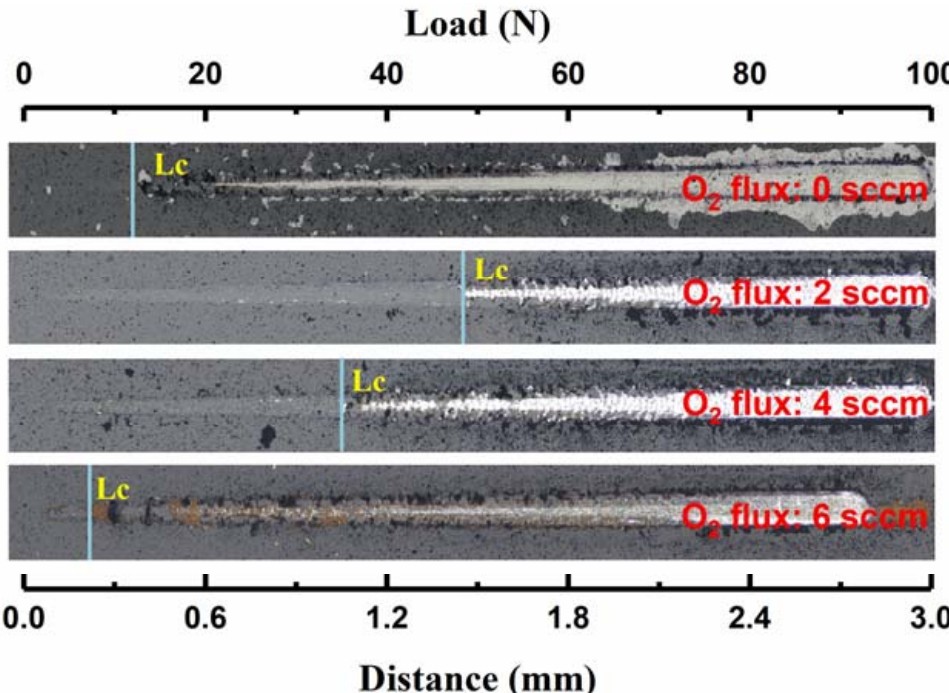

**Figure 7.** Optical morphology of the scratch tracks of the Si/O-DLC coatings deposited with different $O_2$ fluxes.

Figure 8 shows the friction coefficient curves of the Si/O-DLC coatings deposition with different $O_2$ fluxes. The average friction coefficient is calculated from the relative state stage. It can be seen that the pure Si-DLC coating shows a relatively smooth friction process and the average friction coefficient is approximately 0.21. For the coatings deposited with t$O_2$ fluxes of 2 sccm and 4 sccm, the friction coefficient curves become smoother and the average friction coefficient decrease to approximately 0.16. As the $O_2$ flux increases up to

6 sccm, the friction coefficient curves become unstable and the average friction coefficient increases to approximately 0.3. When a certain amount of O is introduced, the silicon oxide compound (e.g., a-SiO$_x$) formed on the surface can effectively reduce the friction coefficient of the Si/O-DLC coating [13]. When the O$_2$ flux is high, however, the mechanical properties of the coating decrease, resulting in an increase in the grinding and friction coefficient during the friction process.

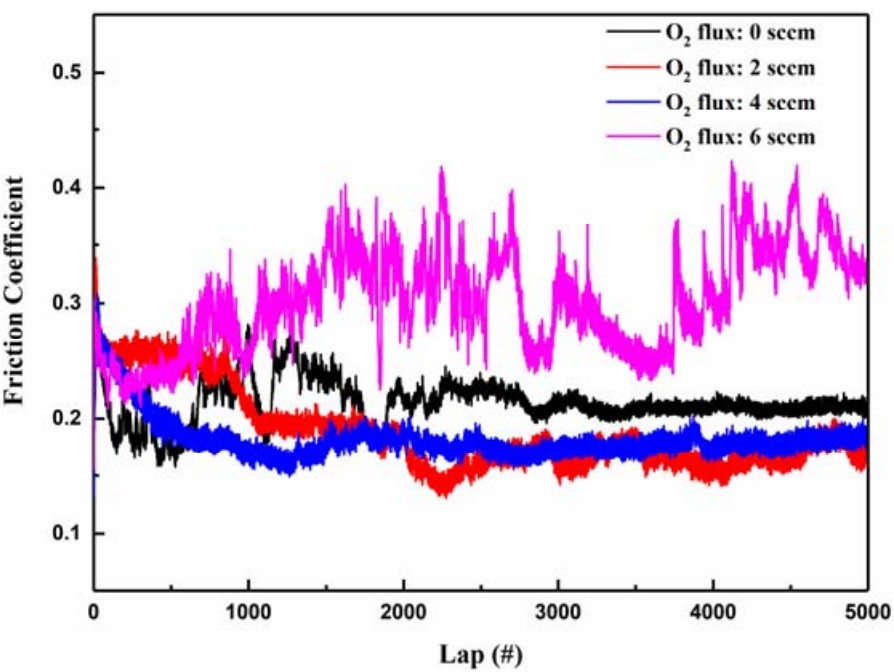

**Figure 8.** Friction coefficient curves of Si/O-DLC coatings deposition with different O$_2$ fluxes.

The surface resistance of the Si/O-DLC coatings is shown in Figure 9 as a function of the O$_2$ flux. It can be seen that the pure Si-DLC coating has an electrical resistivity of approximately $1.77 \times 10^4$ Ω·cm. When the O is doped into the coatings, the surface resistivity of the coatings grows rapidly to $6.16 \times 10^7$ Ω·cm and increases to $9.42 \times 10^8$ Ω·cm when the O$_2$ flux fraction increases to 4 sccm. Generally, the electrical resistivity of the DLC coatings has a significant relationship with the carbon structure. The DLC coatings with a high sp$^2$ content (low sp$^3$ content) possess a low resistivity [3]. The surface resistivity of the hydrogenated DLC (e.g., a-C:H) coatings with the sp$^3$ content (20~60%) is $10^4$–$10^{12}$ Ω·cm. It is reported that the hydrogen-free DLC coatings deposited by sputtering are usually very low in sp$^3$ content. Accordingly, the electrical resistivity of the DLC coatings deposited by sputtering is approximately 10 Ω·cm [31]. Furthermore, when Si and O are co-doped into the DLC coating, the a-SiO$_x$ compound is formed and embedded in the C-C network structure. The insulating a-SiO$_x$ compound hinders electron transport, resulting in an increase in the resistivity of the Si/O-DLC coatings [28]. As the O$_2$ flux increases to 6 sccm, the coatings show a significant increase in the sp$^2$-C fraction, causing a decline in the surface resistivity. However, the insulating a-SiO$_x$ among the sp$^2$-C fraction also effectively blocks electron transport. As a result, the Si/O-DLC coatings also have a high surface resistance. It is believed that we can acquire high-insulating DLC coatings deposited by magnetron sputtering through Si and O co-doping. The Si/O-DLC coatings with good mechanical properties, tribological performance and electrical insulation properties might be used as protection and insulation layers for microelectronics.

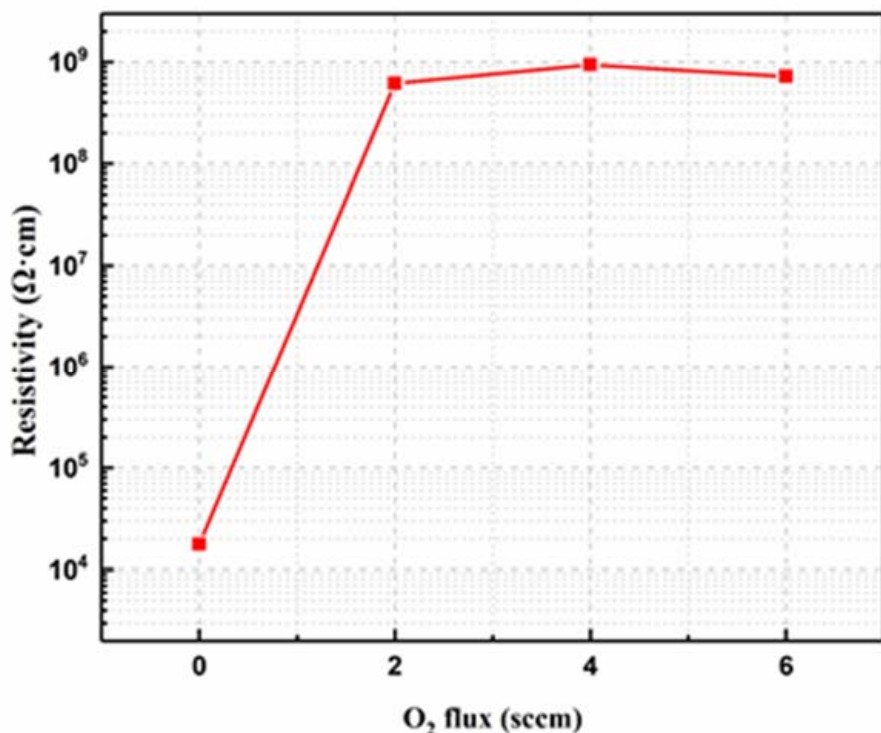

**Figure 9.** The surface resistance of the Si/O-DLC coatings deposition with different $O_2$ fluxes.

## 4. Conclusions

Si/O-DLC coatings were deposited by DC magnetron sputtering technology using $C_2H_2$, $O_2$ and Ar as sputtering gases. The influences of O and Si co-doping on the composition, bond structure, mechanical properties, tribological properties and electrical resistivity of the Si/O-DLC coatings were studied as a function of the $O_2$ flux in the gas mixture. It was found that the doping content of O in the coatings could be controlled by the $O_2$ flux. The doping Si favors bonding with O to form a Si-O-Si bond (silicon-oxygen compound) rather than a Si-C bond, resulting in a decrease in the $sp^3$-C content of the coatings. The hardness and elastic modulus of the coatings decreases with the $O_2$ flux due to the graphitization of the Si/O-DLC coatings. However, a small amount of O doping has little influence on the hardness of the Si/O-DLC coating. It is worth noting that the Si/O-DLC coatings exhibit a higher adhesive strength compared to the pure Si-DLC coatings. Additionally, the Si/O-DLC coatings show a good tribological performance with the friction coefficient due to the lubrication of the silicon–oxygen compound. Furthermore, the Si/O co-doping can significantly improve the surface resistivity of the coatings. The formation of the insulating a-$SiO_x$ embedded in the carbon matrix is believed to be attributed to the improvement in the surface resistivity, which can reach $9.42 \times 10^7$ Ωcm. The Si/O-DLC coatings with good mechanical properties, tribological performance and electrical insulation properties might be used as protection and insulation layers for microelectronics.

**Author Contributions:** Conceptualization, W.D. and Q.W.; methodology, W.D.; investigation, L.W.; writing—original draft preparation, L.W.; writing—review and editing, W.D.; funding acquisition, W.D. All authors have read and agreed to the published version of the manuscript.

**Funding:** This work was funded by the National Natural Science Foundation of Guangdong province (Grant No: 2021A1515011921).

**Data Availability Statement:** The data used to support the findings of this study are available from the corresponding author upon request.

**Conflicts of Interest:** The authors declare no conflict of interest.

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
