# Peer review of "Structure and Property of Diamond-like Carbon Coating with Si and O Co-Doping Deposited by Reactive Magnetron Sputtering"

_jcs, doi:10.3390/jcs7050180_

Round 1

Reviewer 1 Report

The manuscript is devoted to the study of the influence of gas composition in DC magnetron sputtering technology on the properties of hydrogenated Si/O-DLC films deposited on tungsten carbide, silicon and quartz substrates. Films of such composition have been known for decades and are fairly well studied. Their main advantage over diamond-like carbon (DLC) films is higher thermal stability due to Si-O-Si bonds. It was not possible to establish that the authors of the article proposed and implemented something fundamentally new. The samples obtained by them have properties that are far from record-breaking both in terms of hardness and friction coefficient. To determine the composition of the films, the authors used X-ray photoelectron spectroscopy with a step size of 1 eV. The sensitivity of their measurements did not allow to determine separately contribution of sp2/sp3 carbon, as well as the fractions of oxygen- and hydrogen-containing bonds. If it is possible I would like recommend additional measurements of Rutherford backscattering spectrometry, hydrogen forward scattering spectrometry or secondary ion mass spectroscopy.

The IR absorption spectra are given only in relative units, which did not allow the authors to determine the concentration of bound hydrogen in various states. I hope that the authors will make appropriate corrections and discuss in the text of the manuscript not only the effect of silicon and oxygen on the properties of the deposited films, but also of hydrogen in its various states. Or they should honestly admit that the presence of hydrogen in the films was not taken into account due to the method limitations.

There are some questions about measurements. The claimed focusing of the laser beam can lead to thermal annealing of the films. How was this taken into account in the measurements?

Figures 1 and 2 should be interchanged, since plotting in Figure 1 is impossible without the measurements shown in Figure 2.

The accuracy of determining the fraction of carbon (even without taking into account the contribution of hydrogen) is greatly overestimated at the existing quality of X-ray photoelectron spectroscopy measurements.

Authors in Conclusions write about the results of measurements the resistivity and conductivity. Please, explain what is the difference between them.

The main note. The novelty of the work and the place of research should be clearly indicated, at least in terms of a ten-year-old review of similar films - Meškinis, Š., & Tamulevičienė, A. (2011). Structure, properties and applications of diamond like nanocomposite (SiOx containing DLC) films: a review. Materials science, 17(4), 358-370.

Author Response

Kindly check the attachment

Reviewer 2 Report

In jcs-2340013 “Structure and property of diamond-like carbon coating with Si and O co-doping deposited by reactive magnetron sputtering”, authors report on the structure, resistivity and tribological performance of SiOx-DLC material. Nanocarbon composites are widely studied, and the study is relevant for the J.Compos.Sci. The reported variation of the films resistivity within 5 orders of magnitude is quite a prominent result. However, there are several issues that should be resolved.

Major comments:

1) In lines 19 and 281 and within the text you claim that SiOx is "solid-dissolved" in carbon structure. Does that mean that there are somewhat uniformly distributed SiOx groups that don't form any kind of nanoclusters/nanoparticles? Are there any TEM/SAED studies or literature data proving that? In the review [10.5755/j01.ms.17.4.770] it is stated that “DLC:SiOx (diamond like nanocomposite) films are a class of the DLC based composite materials which can consist of the interpenetrating networks of a-C:H and SiOx or SiOx clusters embedded into the a-C:H matrix”, and the “solid solution” is not reported. If there is no proof that the clusters are not formed, I think that the "solid-dissolved" term should be reformulated and more general term (for example, “doping”) should be applied.

2) In FTIR spectra, the feature at ~1600 cm-1 may be attributed to C=C bonds [10.1134/S106377612212010X], therefore, its emergence for the “6 sccm” sample may prove the graphitization observed by Raman spectroscopy. It would be beneficial to supplement the manuscript with the appropriate discussion.

3) Fig. 8 shows that the SiOx doping resulted in the resistivity increase although overall sp2/sp3 ratio raised. Could you please explain/suggest why does the SiOx doping plays more important role in the change of the resistivity than the altering of the carbon matrix?

4) Obtained hardness value of 10 GPa seems like quite a modest result. For example, is significantly surpassed by the pristine ta-C material, for which the 21-24 GPa value was reported [10.1088/1742-6596/1799/1/012027]. Is it possible to compare this result with the values reported for SiOx-DLC and/or to prove that it will be sufficient for the declared protection layers application?

Minor comments:

5) Line 13, please revise the fragment "scratch mete".

6) Is it possible to determine the thickness of the deposited coatings?

7) By "Splicing target" (line 10, 71 etc.), did you mean "spliced target" (see patent CN203112922U "Metal spliced target for preparing film by magnetron sputtering")? The details of the target preparation should be presented in Section 2.

8) Lines 27-35 remaining from the paper template should be removed.

9) By "based pressure" (lines 85-86), did you mean "base pressure"?

10) Lines 145-155: appropriate references should be provided when you ascribe the XPS line positions to the chemical shifts related to various bonds.

11) Lines 171-174: appropriate references should be provided when you ascribe FTIR line positions to the vibrations of the bonds.

12) By "sp3-hybridized Si-C bond", did you mean "Si-C(sp3) bond"? The bond cannot have sp3 hybridization, but carbon atom can.

13) Fig 2: what kind of XPS baseline was subtracted (Shirley/Tougaard)? How were the peaks fitted, what kind of software was used?

14) By "G-peak position increase", did you mean "shift to larger wavenumber" (line 198)?

15) Fig4b, X axis title: in "O2", the number should be in a subscript, not in a superscript mode.

16) In fig. 5, the naming of red bars ("Modulus") seems confusing.

17) In Fig. 6, what does the “Lc2” notation mean?

18) Lines 258-260 are also quite confusing: if the coating is "diamond-like", then it has a high number of sp3 hybridized atoms [10.1126/science.1114577]. If the sample is "low in the sp3 content", then it has graphite-like structure. Please revise the fragment.

19) Line 261-262: could you explain the mechanism of doping-induced formation of C(sp3)-C(sp3) bonds in more detail?

Author Response

Kindly check the attachment

Round 2

Reviewer 1 Report

The authors of the manuscript in its second edition corrected a number of inaccuracies. In general, I am satisfied with their explanations. However, additional changes to the text need to be made.

1. Lines 214-218. As you know, diamond-like carbon films usually contain 20-40 at. % of hydrogen. The XPS method for determining the hydrogen concentration has a low sensitivity; HREELS is better suited for this. In this regard, the authors of the manuscript are not entitled to determine the content of the atomic fractions of silicon, oxygen and carbon in the studied samples only on the basis of XPS measurements (for carbon, it would be better to determine the fraction of sp2 and sp3 of carbon). The signal-to-noise ratio in the Figure 2 allows this to be done with an accuracy of no more than 1%. This accuracy is quite sufficient for the analysis of the obtained results on the mechanical and optical properties of the films. The way out in this situation may be to analyze the ratio of silicon/carbon, oxygen/carbon or silicon/oxygen, as was done in [Ayiania M. et al. Deconvoluting the XPS spectra for nitrogen-doped chars: An analysis from first principles //Carbon 162 (2020) 528].

2. Figure 4. On the vertical axis, the Absorbance value is shown as having no dimension. This does not even allow a qualitative comparison of the spectra of films deposited under different technological conditions. It is proposed to either recalculate these spectra into absorption coefficient spectra, or specify the film thicknesses in the captions. It is also necessary to add a phrase like this - Spectra are shifted vertically one by one for clarity.

Author Response

Kindly check the attachment

Reviewer 2 Report

1) Lines 125-126, "The thickness of the deposited coatings was measured using ball crater tester (CaloTest CAT2c, Anton Paar, Austria) and ranging from about 0.7 mkm to 0.6 mkm as the O2 flux increases from 0 sccm to 6 sccm." Could you please explain why the thickness of the coatings decreases with the increase of the O2 flux?

2) Line 144-145, did you mean that the grating with 500 lines per mm was used alongside with the 10% filter? I don't understand how the 500 nm filter can filter the 633 nm Rayleigh-scattered light of the laser used for Raman spectra excitation.

3) Lines 260-261, "C and H mainly exist in the form of -CH2 functional group": as FTRI is not a quantitative method, this sentence seems misleading.

4) Please comment on the errors of the C1s components positioning associated with the calibration of the spectra to the C1s line position of 284.6 eV. Typically, is is hard to analyse the position of the components when the position of the peak is pre-set.

5) Lines 383-384: please describe how the SiOx formation isolates sp2-hybridized carbon clusters.

Author Response

Kindly check the attachment

Round 3

Reviewer 1 Report

The paper can be accepted in present form

Author Response

We have checked the manuscript carefully. We thank the reviewer for the good comments and advices.

Reviewer 2 Report

Authors of jcs-2340013 “Structure and property of diamond-like carbon coating with Si and O co-doping deposited by reactive magnetron sputtering” have addressed most of my concerns. I appreciate their feedback and the improvements they have made. However, I have a few additional suggestions for them to consider.

11)      Line 129: “ND” abbreviation should be defined.

22)  The explanation presented as a reply to the question 1 (lines 108-111) seems confusing, as argon flux didn’t change throughout the experiment, therefore, according to authors’ logic, the presence of additional oxygen should increase the amount of sputtered material. That’s not the case, however, as denser gas can be less ionized, thus reducing the efficiency of the sputtering. As authors don’t discuss the ionization of the work gas and the efficiency of the sputtering, and the change of the films’ thickness is slight, I suggest that the presented discussion should be removed.

Author Response

(1) we define the ND in the revised manuscript. It is "neutral density"

(2) we remove this sentence.

We thank the reviewer for the nice comments and advices.

Round 4

Reviewer 2 Report

The manuscript was substantially improved throughout the review process. I recommend it to be produced as is.